# Detection of Prosthetic Loosening in Hip and Knee Arthroplasty Using Machine Learning: A Systematic Review and Meta-Analysis

**DOI:** 10.3390/medicina59040782

**Published:** 2023-04-17

**Authors:** Man-Soo Kim, Jae-Jung Kim, Ki-Ho Kang, Jeong-Han Lee, Yong In

**Affiliations:** Department of Orthopaedic Surgery, Seoul St. Mary’s Hospital, College of Medicine, The Catholic University of Korea, 222, Banpo-daero, Seocho-gu, Seoul 06591, Republic of Korea

**Keywords:** loosening, arthroplasty, machine learning, transfer learning, review, prosthesis

## Abstract

*Background*: prosthetic loosening after hip and knee arthroplasty is one of the most common causes of joint arthroplasty failure and revision surgery. Diagnosis of prosthetic loosening is a difficult problem and, in many cases, loosening is not clearly diagnosed until accurately confirmed during surgery. The purpose of this study is to conduct a systematic review and meta-analysis to demonstrate the analysis and performance of machine learning in diagnosing prosthetic loosening after total hip arthroplasty (THA) and total knee arthroplasty (TKA). *Materials and Methods*: three comprehensive databases, including MEDLINE, EMBASE, and the Cochrane Library, were searched for studies that evaluated the detection accuracy of loosening around arthroplasty implants using machine learning. Data extraction, risk of bias assessment, and meta-analysis were performed. *Results*: five studies were included in the meta-analysis. All studies were retrospective studies. In total, data from 2013 patients with 3236 images were assessed; these data involved 2442 cases (75.5%) with THAs and 794 cases (24.5%) with TKAs. The most common and best-performing machine learning algorithm was DenseNet. In one study, a novel stacking approach using a random forest showed similar performance to DenseNet. The pooled sensitivity across studies was 0.92 (95% CI 0.84–0.97), the pooled specificity was 0.95 (95% CI 0.93–0.96), and the pooled diagnostic odds ratio was 194.09 (95% CI 61.60–611.57). The I2 statistics for sensitivity and specificity were 96% and 62%, respectively, showing that there was significant heterogeneity. The summary receiver operating characteristics curve indicated the sensitivity and specificity, as did the prediction regions, with an AUC of 0.9853. *Conclusions*: the performance of machine learning using plain radiography showed promising results with good accuracy, sensitivity, and specificity in the detection of loosening around THAs and TKAs. Machine learning can be incorporated into prosthetic loosening screening programs.

## 1. Introduction

Total hip arthroplasty (THA) and total knee arthroplasty (TKA) are effective procedures that significantly improve quality of life and functional recovery in orthopedic surgery [1,2,3,4,5,6,7,8]. THA and TKA show sufficient survival rates [9,10], with 95% survivorship over 15 years in the case of TKA [10] and over 90% survivorship over 10 years in the case of THA, with continuous technological advancement [9]. Due to these successful results, the use of THAs and TKAs is continuously increasing. However, the ratio of revision THA and TKA is also expected to continue increasing because of continuous increases in total joint arthroplasty surgery and average life expectancy [11,12,13,14,15]. In the United States, revision TKA is expected to increase by 601% between 2005 and 2030; revision THA is expected to increase by 137% [13].

As revision THAs and TKAs increase, the associated economic burden is expected to gradually increase [16,17]. Therefore, understanding the causes and risk factors of revision THA and TKA to improve the durability of revision surgery is increasingly important [16,17]. The most common causes of revision TKA are infection and aseptic loosening; the rate of aseptic loosening has increased recently [16,17,18,19]. In revision THA, infection and aseptic loosening also account for the largest proportion of revision surgeries [20,21,22]. As the diagnosis of prosthetic loosening is still challenging, various imaging tools are used for diagnosis, including plain radiographs, scintigraphy, arthrograms, fluorodeoxyglucose-positron emission tomography (FDG-PET) scans, and MRI [23,24]. However, except for plain radiography, these tools are invasive and expensive; therefore, plain radiography is the most cost effective method [25]. In addition, there is diversity in terms of concordance rates among the expert physicians assessing the cases [26,27].

Due to recent developments, machine learning has begun to be used widely in orthopedic surgery [28,29]. In particular, machine learning shows excellent performance in the field of diagnosis through images [28,29]. In fact, various techniques using machine learning are being used to diagnose lung disease [30] and breast cancer using radiographs [31]; these techniques are actually used as auxiliary diagnostic devices to assist doctors in diagnosis in hospitals [30,31]. Studies using machine learning to detect prosthetic loosening after arthroplasty surgery in orthopedic surgery are also continuously increasing in number [32,33,34,35,36]. Therefore, the purpose of this study is to conduct a systematic review and meta-analysis to demonstrate the analysis and performance of machine learning in diagnosing prosthetic loosening after total hip arthroplasty (THA) and total knee arthroplasty (TKA). We hypothesized that the model using machine learning is useful and can be incorporated into screening programs in diagnosing prosthetic loosening of THAs and TKAs.

## 2. Materials and Methods

This study was performed following the guidelines of the Preferred Reporting Items for Systematic Reviews and Meta-Analysis (PRISMA) statement (S1 PRISMA Checklist) [37].

### 2.1. Data and Literature Sources

The study design was performed according to the Cochrane Review Methods. Multiple comprehensive databases, including MEDLINE, EMBASE, and the Cochrane Library, were searched in September 2022 for studies in English that detected the loosening of implants using machine learning (S1 Search Strategy). There were no restrictions on publication year. Search terms included mesh “Arthroplasty” and key words “replacement” “joint replacement” “alloarthroplasty”, mesh “machine learning” and key words “transfer learning”, “artificial intelligence”, “deep learning”, “neural network”, “decision trees”, and mesh “osteolysis” and key words “bone loss”, “bone resorption”, “loosening”, “failure”. After the initial electronic search, manual searches of the reference lists and the bibliographies of identified articles, including relevant reviews and meta-analyses, were conducted to identify trials that the electronic search may have missed. Identified articles were individually assessed for inclusion.

### 2.2. Study Selection

Two reviewers independently determined study inclusion according to the pre-defined selection criteria. Titles and abstracts were screened for relevance. In cases of uncertainty, the full article was evaluated to determine eligibility. Discrepancies were resolved through discussion. Studies included met these criteria: (1) used a machine learning algorithm as an index for the diagnosis of prosthetic loosening or osteolysis; and (2) the integrated data (true positive, false negative, false positive, and true negative) were provided directly or indirectly. The exclusion criteria included: (1) animal studies; (2) studies with incomplete data; and (3) reviews, comments, letters, and research for which full text cannot be obtained.

### 2.3. Data Extraction

Two reviewers independently extracted data from each study using a standardized data extraction form. Disagreements were resolved by discussion; those unresolved through discussion were reviewed by a third reviewer. The following variables were included: first author, publication year, country, study type, index test, eligibility criteria, reference standard, type of arthroplasty, sample size, machine learning algorithm, preprocessing, augmentation, model structure, the calculated area under the curve (AUC), accuracy, sensitivity, and specificity. We attempted to contact the study authors for supplementary information when there were insufficient or missing data in the articles. The third senior investigator was consulted to resolve any disagreement during data extraction. 

The quality of all literature was evaluated by two researchers using the Quality Assessment of Diagnostic Accuracy Studies (QUADAS-2) [38], which is composed of patient selection, index test, reference standard, and flow and timing. If there is any disagreement in this process, the third author was responsible for making the decision.

### 2.4. Statistical Analyses

All extracted data analyses and picture production were performed with the R language using R studio. A bivariate random effect model was selected to analyze the true positive, false negative, false positive, and true negative values of 2 × 2 tables recorded in the sheet and test the heterogeneity. The sensitivity, specificity, positive likelihood ratio (PLR), negative likelihood ratio (NLR), diagnostic score, and diagnostic odds ratio (DOR) were calculated after integration. In addition, by drawing the summary receiver operating characteristics (SROC) through the Midas command, AUC discriminated the diagnostic ability of machine learning [39]. 

Heterogeneity was determined using the I2 statistic, with values of 25%, 50%, and 75% considered as indicating low, moderate and high heterogeneity, respectively. Due to the high levels of heterogeneity, a random-effects model was used to combine available data by meta-analysis. Random-effect DerSimonian and Laird models were used to calculate weighted averages of the transformed values, which were then back-transformed to produce final pooled rates [40]. Pre-planned sub-groups were designed according to the type of arthroplasty. Publication bias determination was usually only performed when the number of included articles was 10 or more; however, we used a funnel plot to evaluate publication bias.

## 3. Results

### 3.1. Identification of Studies

A study-flow diagram shows the process for study identification, inclusion, and exclusion (Figure 1). An initial electronic search yielded 1306 studies. Three additional publications were obtained through manual searching. In total, 82 potentially eligible studies were assessed for inclusion after screening titles and abstracts. After we reviewed the full texts, an additional 77 studies were excluded. Finally, five studies were included in the meta-analysis.

### 3.2. Study Characteristics and Quality of Included Studies

The study characteristics are summarized in Table 1. All studies were published from 2019 to 2022. All studies were retrospective studies. In total, 2013 patients with 3236 images were assessed, involving 2442 cases (75.5%) with THAs and 794 cases (24.5%) with TKAs. Among THAs, loosening occurred in 1136 cases (46.5%). Loosening occurred in 343 TKA cases (43.2%). Three studies included only THAs, one study included only TKAs, and one study included both THAs and TKAs. All five studies of these studies were retrospective studies and used the transfer learning method. The most common and best-performing machine learning algorithm was DenseNet; in one study, a novel stacking approach using a random forest showed similar performance to DenseNet. The characteristics of the included studies are summarized in Table 1 and Table 2.

### 3.3. Descriptive Statistics

The study with the highest accuracy was performed by Loppini et al. [34] in 2022 (96.8%), while the study carried out by Borjali et al. [32] in 2019 had the lowest accuracy (77.0%). The specificity and sensitivity values of each machine learning model are presented in Figure 2. Sensitivity values for machine learning models in this study ranged between 0.71 (95% CI: 0.60–0.80) and 0.97 (95% CI: 0.95–0.98) in THA. The pooled sensitivity across studies was 0.93 (95% CI: 0.81–0.98) in THA. Sensitivity values for machine learning models ranged between 0.70 (95% CI: 0.61–0.78) and 0.96 (95% CI: 0.93–0.98) in TKA. The pooled sensitivity across studies was 0.88 (95% CI: 0.81–0.98) in TKA. Specificity values for machine learning models in this study ranged between 0.95 (95% CI: 0.91–0.97) and 0.97 (95% CI: 0.95–0.98) in THA. The pooled sensitivity across studies was 0.96 (95% CI: 0.95–0.97) in THA. Specificity values for machine learning models ranged between 0.91 (95% CI: 0.87–0.94) and 0.95 (95% CI: 0.91–0.97) in TKA. The pooled specificity across studies was 0.93 (95% CI: 0.90–0.95) in TKA. The pooled sensitivity across studies was 0.92 (95% CI: 0.84–0.97), the pooled specificity was 0.95 (95% CI: 0.93–0.96) (Figure 2). 

The pooled positive LR was 16.63 (95% CI: 10.55–26.20), while the pooled negative LR was 0.09 (95% CI: 0.03–0.27). The DOR values in this study ranged between 41.83 (95% CI: 20.51–85.30) and 863.23 (95% CI: 518.14–1438.14) in THA. The pooled DOR across studies was 282.93 (95% CI: 59.31–1349.64) in THA. The DOR values for machine learning models ranged between 43.85 (95% CI: 21.60–89.03) and 251.04 (95% CI: 108.70–579.74) in TKA. The pooled DOR across studies was 103.37 (95% CI: 18.70–571.35) in TKA. The pooled DOR was 194.09 (95% CI: 61.60–611.57) (Figure 3). The I2 statistics for sensitivity and specificity were 96.0% and 62.0%, showing that there was significant heterogeneity. The SROC curve indicated the sensitivity and specificity, as well as the prediction regions, with an AUC of 0.9853 (Figure 4).

### 3.4. Quality Assessment and Publication Biases

The quality assessment results of five studies performed using the QUADAS-2 scale are indicated in Figure 2. The figure shows that the overall quality of the included studies was good; all studies were “unclear” or “low risk” with no study including “high risks”. Although the sample size of the included literature was relatively small, the quality of the research was persuasive. (Figure 5) In addition, the funnel plot was asymmetrical, indicating the tendency of publication bias in this meta-analysis. (Figure 6).

## 4. Discussion

As a result of this systematic review, the diagnostic accuracy of the simple radiographic image-based artificial intelligence (AI) model for discriminating relaxation around implants after THA and TKA was over 0.9 in combined sensitivity, specificity, and AUC. As the machine learning for this purpose is in its immature stage, sufficient research results have not been secured. AI model studies using this imaging have not provided clear conclusions about clinical implementation and widespread use.

In the field of orthopedic surgery, THA and TKA are well known as the most effective and satisfactory surgical treatments [7,8]. In addition, long-term follow-up results have shown excellent survival rates [9,10]. The development of materials and technologies related to THA and TKA has contributed to the increase in the number of THA and TKA worldwide [41,42]. However, artificial joints inevitably have the risk of revision operation; thus, the most common cause of revision operation is loosening due to osteolysis [16,17]. Therefore, there has been a continuous demand for the early detection of prosthetic loosening around the implant. However, the accuracy of detection of prosthetic loosening in practice has not been as high as expected [43,44]. Temmerman et al. [44] analyzed the accuracy using radiography to diagnose cementless femoral component loosening; the sensitivity was 50% and the specificity was 89.5%. Cheung et al. [43] reported sensitivity and specificity of 83% and 82%, respectively. In addition to this, the inter-observer agreement between observers evaluating prosthetic loosening was also surprisingly poor [44]. Therefore, there has been a growing interest in tools that can have constant and high sensitivity, specificity, and accuracy in evaluating prosthetic loosening in THA and TKA [32,33]. Machine learning (ML) is on the rise as an alternative [32,33].

ML is a branch of AI that uses data to learn and improve tasks using various systems and algorithms [45,46]. Deep learning (DL) is a type of ML that allows complex tasks to be learned using large amounts of training information [47]. DL uses artificial neural networks (ANNs) made up of neurons arranged in a hierarchical structure. Convolutional neural networks (CNNs) are a subtype of DL that are effective and excellent at image processing. [48] A CNN uses a complex set of layers through which data is passed with a filter that can be trained to create a final or output layer [48]. The studies investigated in this review all have a common feature of using one of the various CNN algorithms [32,33,34,35,36]. This study is significant in its being the first to review studies that differentiate loosening around artificial joints using a CNN model. That model demonstrates excellent ability for image discrimination through plain radiographic images in the field of orthopedics [32,33,34,35,36].

Studies using AI in relation to arthroplasty in orthopedic surgery are gradually increasing in number [28,29]. These studies involve prediction of arthroplasty component size [49], length of stay and costs before primary arthroplasty [50], transfusion after arthroplasty [51], patient dissatisfaction following primary arthroplasty [52], and automated detection and classification of arthroplasty implant from knee radiograph [53]. In studies related to images, CNN models are generally most common; all the studies in this review detected prosthetic loosening using widely used CNN models learned through Imagenet. A further two studies used DenseNet [32,34] and one study used Exception [33]. The other two studies analyzed various CNN models and selected the model with the highest accuracy, which was DenseNet [35,36]. One study had limitations in accuracy with only images [36] while increasing the accuracy to more than 90% by providing additional patient data [36]. The remaining studies achieved satisfactory accuracy without additional patient data [32,33,34,35]. CNN models have been and are continuing to be developed. Additional research using the new CNN models will be needed in the future.

All five studies showed satisfactory results in terms of accuracy, sensitivity and specificity [32,33,34,35,36]. Accuracy, sensitivity, and specificity all showed high values of 0.9 or more [32,33,34,35,36]. The accuracy shown using simple plain radiographs, rather than more advanced imaging, shows similar or superior results compared to studies conducted only with imaging in other medical fields [28]. In a study by Adams et al., deep CNN was used for diagnosing femur neck fractures. The studies in this review showed excellent results with accuracy, specificity, and sensitivity of about 0.9 [54]. In a study by Urakawa et al., hip intertrochanteric fracture was diagnosed using a deep CNN model; the results of that study also showed a similar 95% accuracy [55]. In fact, two studies in this review compared diagnosis rates with orthopedic surgeons and showed higher accuracy than orthopedic surgeons [32,33]. Lau et al. reported that the accuracy of the machine learning model was 96.3%, while the average of the two orthopedic surgeons was 92% [33]. Borjali et al. demonstrated that the orthopedic surgeons’ accuracy were 77%, which was lower than that of the machine learning model [32]. In particular, the specificity was similar; however, the sensitivity was poor in that study, indicating that loosening was not well differentiated [32].

Even when examining THA and TKA separately by sub-group analysis, the machine learning model showed high accuracy, sensitivity, and specificity in discriminating loosening around THAs and TKAs [32,33,34,35,36]. In the case of THA, four studies were included and showed excellent results, with accuracy, sensitivity, and specificity above 90% [32,34,35,36]. In the case of TKA, there are limitations as only data from two studies were included; however, this result also showed over 90% accuracy, sensitivity, and specificity [33,36]. Since the number of TKA samples was too small to perform sub-group analysis according to the type of joint arthroplasty, clear results require additional future TKA studies [33,36].

In the results of the meta-analysis, the heterogeneity among studies was particularly high at over 90% in sensitivity and, except in some cases, high in specificity [32,33,34,35,36]. Although difficulty arises in clearly explaining such high heterogeneity, potential reasons for heterogeneity include the diversity of CNN models used in this study, differences in patients’ demographic data, differences in processes in creating CNN models, and the number of images used in the study [32,33,34,35,36]. This can be inferred from differences in quality. In this study, a random effect model was used due to high heterogeneity; this model was high in all values, including sensitivity and specificity.

There were some limitations in our research. Firstly, this study only included five articles; thus, the sample size was relatively small with only 1000 cases in the loosening group and 1500 cases in the non-loosening group [32,33,34,35,36]. Secondly, though four of the five studies contained data collected by the researchers at the hospital [32,33,34,36], one study contained image data collected from published papers available on the Internet [35]. Therefore, future research should focus on using high-quality data collected from hospitals or research centers in a broader population of patients with periprosthetic osteolysis or loosening with various clinical symptoms. Furthermore, future studies should explicitly describe the role of machine learning tools being developed as screening, diagnostic, or prognostic tools. A third limitation is that publication bias may exist. One possible cause is that there may be unpublished studies using underperforming models. In addition, the overall model of this study was highly heterogeneous and included a significant number of studies that could contribute to both the occurrence of publication bias and the high statistical power of the asymmetric test. Future meta-analyses should ensure conduct that includes pre-print articles that can reduce publication bias. Fourthly, the language restrictions in the included studies may have increased the risk of bias in the study results. Fifthly, there are many different machine learning models available with different variants and parameters. However, we were unable to compare each variant of the model due to the insufficient sample size of individual models. Lastly, most studies developed new models for TKA loosening detection but did not perform external validation using other data sources. One common concern is that the local data sets used for validation may not be representative of the target population on a global scale. In future studies, external validation requires improvement.

This meta-analysis is of substantial significance in that this is the first method to quantitatively combine and interpret data examining prosthetic loosening from different studies, potentially providing key clues for clinical application and further research. This study provided an opportunity to examine the accuracy of the method using ML in the detection of prosthetic loosening in THA and TKA. In addition, it was confirmed that the model using ML could be used as an auxiliary aid device in detecting prosthetic loosening in clinical practice. However, additional large-scale analysis studies and more research would be needed.

## 5. Conclusions

The performance of machine learning on plain radiographs showed promising results with good accuracy, sensitivity, and specificity in the detection of loosening around THAs and TKAs. Machine learning can be incorporated into prosthetic loosening screening programs. However, more research results are needed to clearly judge the ability of the machine learning model to discriminate loosening around the joint arthroplasty.

## Figures and Tables

**Figure 1 medicina-59-00782-f001:**
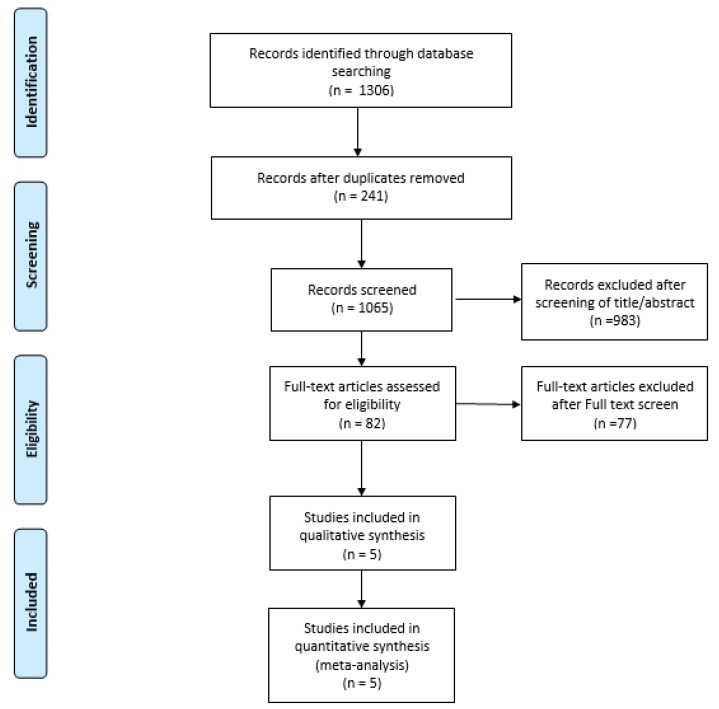
PRISMA flow diagram for systematic review.

**Figure 2 medicina-59-00782-f002:**
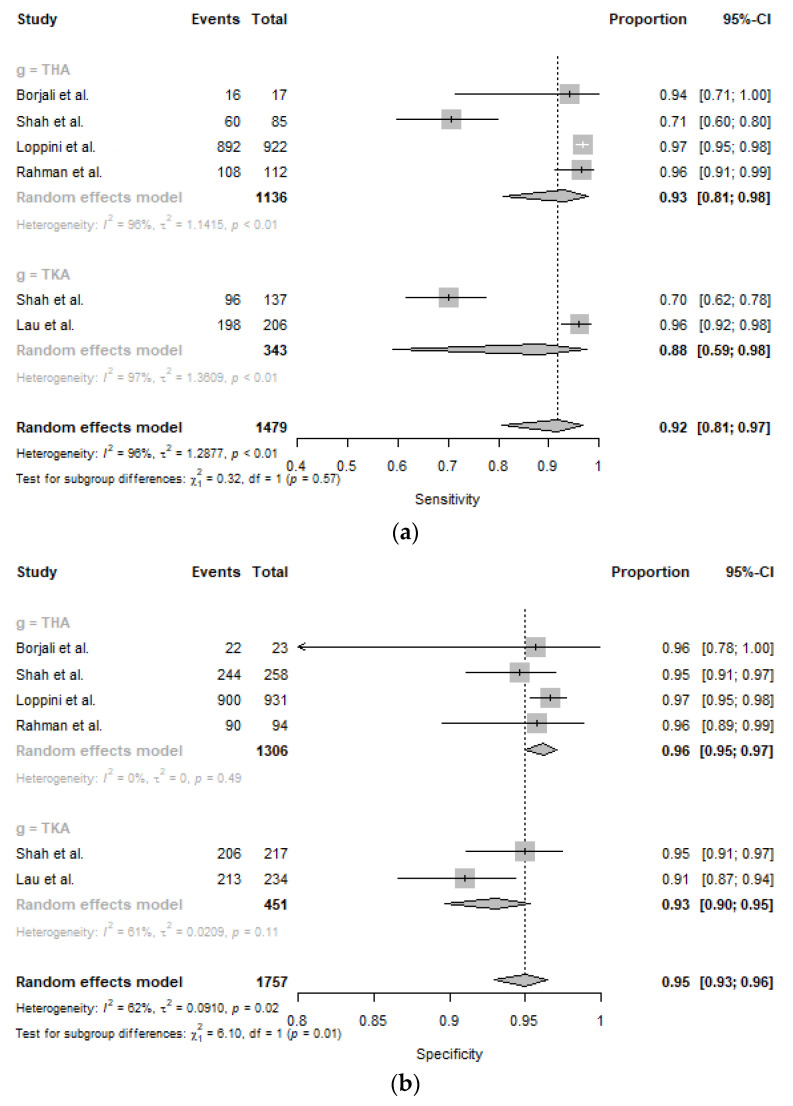
Forest plots for sensitivity (**a**) and specificity (**b**). GLMM: generalized linear mixed model, CI: confidence interval, THA: total hip arthroplasty, TKA: total knee arthroplasty [32,33,34,35,36].

**Figure 3 medicina-59-00782-f003:**
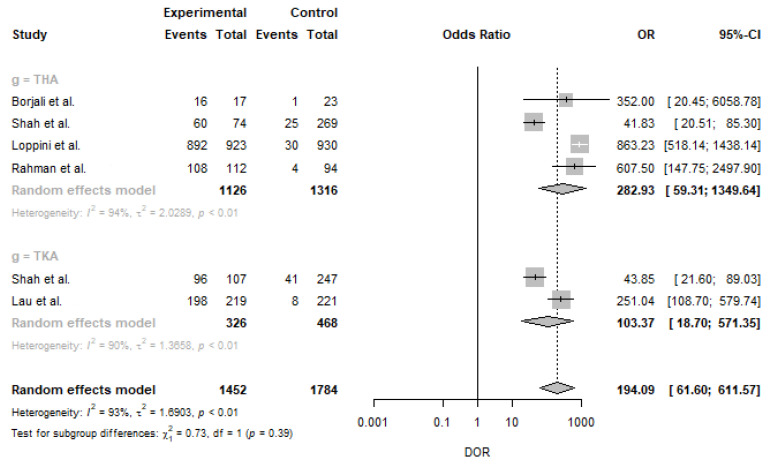
Forest plots diagnostic odds ratio. IV: interval variance; CI: confidence interval, THA: total hip arthroplasty; TKA: total knee arthroplasty; DOR: diagnostic odds ratio [32,33,34,35,36].

**Figure 4 medicina-59-00782-f004:**
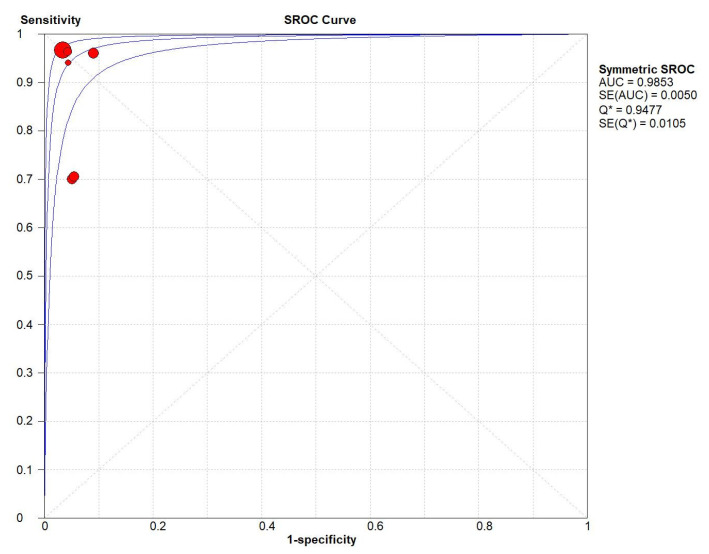
Summary receiver operating characteristics (sROC) curve, the calculated area under the curve (AUC) = 0.985. SROC: summary receiver operating characteristics; SE: standard error; AUC: Area under the curve.

**Figure 5 medicina-59-00782-f005:**
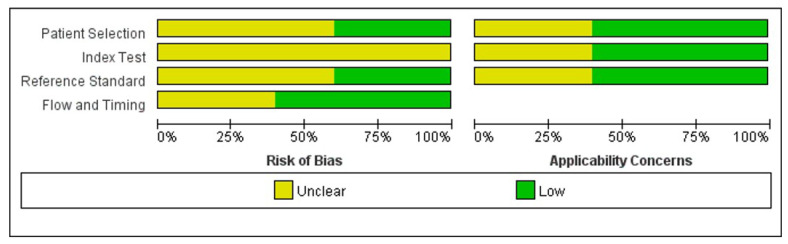
Methodological assessment by QUADAS-2 [32,33,34,35,36].

**Figure 6 medicina-59-00782-f006:**
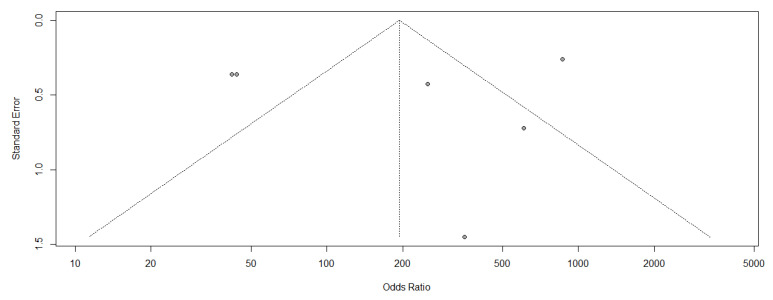
Funnel plot.

**Table 1 medicina-59-00782-t001:** Summary of general study characteristics.

Study	Country	Design	Index Test	Eligibility Criteria	Reference Standard	Arthroplasty Type	Failure Detection	Number of Loosening vs. Non-Loosening	Type of Validation	Data Source
Borjali et al., 2019 [32]	USA	Retrospective study	X-ray	Yes	Operation record	THA	Loosening	17 THAs vs. 23 THAs	Five-fold cross validation	Single center
Shah et al., 2020 [36]	USA	Retrospective study	X-ray Demographic & comorbidity data	Yes	Operation record	THA TKA	Loosening	137 TKAs, 85 THAs vs. 217 TKAs. 258 THAs	Training 60% Validation 20% Test 20%	Single center from 2012–2018
Loppini et al., 2022 [34]	Italy	Retrospective study	X-ray	Yes	Operation record	THA	Loosening malposition wear infection	420 failed THA vs. 210 normal THA 922 failed images vs. 931 non-failed images	Training 63% Validation 27% Test 10%	Single center from 2009–2019
Lau et al., 2022 [33]	Hong Kong	Retrospective study	X-ray Clinical information	Yes	Operation record	TKA	Loosening	206 TKAs vs. 234 TKAs	Test 75% (345 images) Validation 25% (95 images)	Single center
Rahman et al., 2022 [35]	Qatar	Retrospective study	X-ray	Yes	Research results	THA	Loosening	112 THAs vs. 94 THAs	Five-fold cross validation Training 70% Validation 10% Test 20%	Images from published article

THA: total hip arthroplasty; TKA: total knee arthroplasty.

**Table 2 medicina-59-00782-t002:** Artificial intelligence-based prediction model characteristics described in included studies.

Study	AI Method	Pre-Processing	Augmentations	Model Structure	AUC	Accuracy	Sensitivity	Specificity	AI vs. Expert Doctor
Borjali et al., 2019 [32]	DL	Transfer learning	Reorientation, magnification	DenseNet Re-trained CNN Pre-trained CNN	Pre-trained 0.950 Re-trained 0.800	Pre-trained 0.950	Pre-trained 0.940	Pre-trained 0.960	Orthopaedic surgeon; accuracy 0.770 sensitivity 0.530 specificity 0.960
Shah et al., 2020 [36]	DL	Resize segmentation Ttransfer learning	None	ResNet AlexNet Inception DenseNet		Resnet 0.882 Alexnet 0.901 Inception 0.922 DenseNet 0.953 Best-model overall 0.883 TKA 0.858 THA 0.901	Best model overall 0.702 TKA 0.698 THA 0.703	Best model overall 0.956 TKA 0.952 THA 0.946	None
Loppini et al., 2022 [34]	DL	Resize transfer learning	Transformation Horizontal flip Rotation Zoom	DenseNet	0.993	Training 0.990 Validation 0.975 Test 0.968	0.968	0.968	None
Lau et al., 2022 [33]	DL	Transfer learning	None	Xception	Pre-trained test 0.935	0.963	0.961	0.909	Two senior orthopaedic specialists with 15–20 years’ experience; accuracy 0.921
Rahman et al., 2022 [35]	DL	Cropping resize normalization transfer-learning	Rotation Scaling translation	Resnet18 Resnet50 Resnet101 InceptionV3 DenseNet161 DenseNet201 Mobilentetv2 Googlenet Staking approach	DenseNet201 Staking approach using Random forest	DenseNet 0.947 Random forest 0.961	DenseNet 0.9467 Random forest 0.964	DenseNet 0.945 Random forest 0.964	None

AI: artificial intelligence, AUC: area under the curve, DL: deep learning, CNN: convolutional neural networks, THA: total hip arthroplasty, TKA: total knee arthroplasty.

## Data Availability

The data presented in this study are available in the main article.

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
