# Peer review of "Detection of Prosthetic Loosening in Hip and Knee Arthroplasty Using Machine Learning: A Systematic Review and Meta-Analysis"

_medicina, 2023, doi:10.3390/medicina59040782_

Round 1
Reviewer 1 Report
This is a well performed meta-analysis on the diagnosis of periprosthetic loosening by machine learning in THA and TKA. This analysis incorparates five retrospective studies. The introduction is concise with a clear aim. The methods are appropriate according to the PRISMA guidelines. The results are presented in a clear way. The discussion is thorough and incorporates recent and relevant literature. Overall I can recommend publication of this paper.
Author Response
â–¶We thank the reviewer for his/her valuable time and we agree with this succinct summary of our study.
Reviewer 2 Report
Dear authors,
Thank you very much for your work! Even if the number of identified studies is very low your review demonstrates that machine learning could be a very useful tool to identify aseptic loosening in THA an TKA in the future.
best wishes,
Clemens Kösters
Author Response
We thank the reviewer for his/her valuable time and we agree with this succinct summary of our study.
Reviewer 3 Report
Dear author,
I am pleased to submit to you my review of your article.
The topic is interesting, current, and relevant to our clinical practice.
The article is well written, but many concerns burden it with a minor revision before it can be accepted for publication. After these minor revisions, in my opinion, it is publishable. Congratulations to the authors for their efforts.
Some suggested corrections have been listed below. Please answer point by point.
INTRODUCTION
-On page 2, lines 46-48, at the end of the sentence, "The most common causes…has increased recently." I suggest including recent references that mention the clinical importance of specific conditions that lead to revision surgery in TKA, such as infections and the need to perform in-one and two-stage revisions and aseptic loosening that are often related to polyethylene wear. I recommend you add these related references: doi: 10.1007/s00590-023-03480-7; doi: 10.1007/s00167-022-07226-6.
-On page 2, lines 46-48, at the end of the sentence, "In revision THA…largest proportion of revision surgeries." I suggest including recent references that mention the clinical importance of specific conditions that lead to revision surgery in THA, such as infections and the need to perform in-one and two-stage revisions and aseptic loosening that are often related to polyethylene wear. I recommend you add these related references: doi: 10.1016/j.jhin.2022.05.002;
doi: 10.1177/11207000221140346.
TABLES AND FIGURES:
Add a legend where you enter the abbreviations given.
DISCUSSION
-On page 3, line 188, this sentence is the aim already described in the Introduction and should be removed from here. Insert the main finding of the study instead. Always start the discussion by writing the main finding of the study.
-Respect abbreviations. On page 3, Line 189: The first time you mention "AI," you should write "Artificial intelligence (AI)". There is some confusion. After mentioning an acronym for the first time, you can use only the abbreviation instead of writing the whole word. Correct the abbreviations in the text.
- On page 3, lines 188-190, at the end of the sentence, add recent references that mention Artificial Intelligence. I recommend you add these related references: doi: 10.1016/j.artd.2023.101116.
-At the end of the discussion section, add the study's strengths, which are essential to a high-quality written manuscript.
Author Response
Dear author,
I am pleased to submit to you my review of your article.
The topic is interesting, current, and relevant to our clinical practice.
The article is well written, but many concerns burden it with a minor revision before it can be accepted for publication. After these minor revisions, in my opinion, it is publishable. Congratulations to the authors for their efforts.
â–¶We thank the reviewer for his/her valuable time and we agree with this succinct summary of our study.
Some suggested corrections have been listed below. Please answer point by point.
INTRODUCTION
-On page 2, lines 46-48, at the end of the sentence, "The most common causes…has increased recently." I suggest including recent references that mention the clinical importance of specific conditions that lead to revision surgery in TKA, such as infections and the need to perform in-one and two-stage revisions and aseptic loosening that are often related to polyethylene wear. I recommend you add these related references: doi: 10.1007/s00590-023-03480-7; doi: 10.1007/s00167-022-07226-6.
â–¶Thank you for your comments. We added studies presented by reviewers as references. (Line 46)
-On page 2, lines 46-48, at the end of the sentence, "In revision THA…largest proportion of revision surgeries." I suggest including recent references that mention the clinical importance of specific conditions that lead to revision surgery in THA, such as infections and the need to perform in-one and two-stage revisions and aseptic loosening that are often related to polyethylene wear. I recommend you add these related references: doi: 10.1016/j.jhin.2022.05.002; doi: 10.1177/11207000221140346.
â–¶Thank you for your comments. We added studies presented by reviewers as references. (Line 48)
TABLES AND FIGURES:
Add a legend where you enter the abbreviations given.
â–¶Thank you for your comments. We added the legend of the abbreviations. (Tables 1, 2 and Figures 2, 3, 4)
DISCUSSION
-On page 3, line 188, this sentence is the aim already described in the Introduction and should be removed from here. Insert the main finding of the study instead. Always start the discussion by writing the main finding of the study.
â–¶Thank you for your comments. We deleted the aim and insert the main finding of the study in the discussion. (Lines 196-204)
-Respect abbreviations. On page 3, Line 189: The first time you mention "AI," you should write "Artificial intelligence (AI)". There is some confusion. After mentioning an acronym for the first time, you can use only the abbreviation instead of writing the whole word. Correct the abbreviations in the text.
â–¶Thank you for your comments. We correct the abbreviations in the text. (Line 197)
- On page 3, lines 188-190, at the end of the sentence, add recent references that mention Artificial Intelligence. I recommend you add these related references: doi: 10.1016/j.artd.2023.101116.
â–¶Thank you for your comments. We added studies presented by reviewers as references. (Line 210)
-At the end of the discussion section, add the study's strengths, which are essential to a high-quality written manuscript.
â–¶Thank you for your comments. We added the study’s strengths at the end of the discussion section. (Lines 292-294)